# *Acinetobacter baumannii* K106 and K112: Two Structurally and Genetically Related 6-Deoxy-l-talose-Containing Capsular Polysaccharides

**DOI:** 10.3390/ijms22115641

**Published:** 2021-05-26

**Authors:** Anastasiya A. Kasimova, Nikolay P. Arbatsky, Jacob Tickner, Johanna J. Kenyon, Ruth M. Hall, Michael M. Shneider, Alina A. Dzhaparova, Alexander S. Shashkov, Alexander O. Chizhov, Anastasiya V. Popova, Yuriy A. Knirel

**Affiliations:** 1N.D. Zelinsky Institute of Organic Chemistry, Russian Academy of Sciences, 119991 Moscow, Russia; nastia-kasimova979797@mail.ru (A.A.K.); nikolay.arbatsky@gmail.com (N.P.A.); alina98d@gmail.com (A.A.D.); shash@ioc.ac.ru (A.S.S.); yknirel@gmail.com (Y.A.K.); 2Centre for Immunology and Infection Control, School of Biomedical Sciences, Faculty of Health, Queensland University of Technology, Brisbane, QLD 4059, Australia; j.tickner@uq.edu.au (J.T.); johanna.kenyon@qut.edu.au (J.J.K.); 3School of Life and Environmental Sciences, The University of Sydney, Sydney, NSW 2006, Australia; ruth.hall@sydney.edu.au; 4M. M. Shemyakin and Yu. A. Ovchinnikov Institute of Bioorganic Chemistry, Russian Academy of Sciences, 117997 Moscow, Russia; mm_shn@mail.ru; 5State Research Center for Applied Microbiology and Biotechnology, Obolensk, 142279 Moscow Region, Russia; popova_nastya86@mail.ru

**Keywords:** *Acinetobacter baumannii*, capsular polysaccharide, K locus, K106, K112, 6-deoxy-l-talose

## Abstract

Whole genome sequences of two *Acinetobacter baumannii* clinical isolates, 48-1789 and MAR24, revealed that they carry the KL106 and KL112 capsular polysaccharide (CPS) biosynthesis gene clusters, respectively, at the chromosomal K locus. The KL106 and KL112 gene clusters are related to the previously described KL11 and KL83 gene clusters, sharing genes for the synthesis of l-rhamnose (l-Rha*p*) and 6-deoxy-l-talose (l-6dTal*p*). CPS material isolated from 48-1789 and MAR24 was studied by sugar analysis and Smith degradation along with one- and two-dimensional 1H and 13C NMR spectroscopy. The structures of K106 and K112 oligosaccharide repeats (K units) l-6dTal*p*-(1→3)-D-Glc*p*NAc tetrasaccharide fragment share the responsible genes in the respective gene clusters. The K106 and K83 CPSs also have the same linkage between K units. The KL112 cluster includes an additional glycosyltransferase gene, Gtr183, and the K112 unit includes α l-Rha*p* side chain that is not found in the K106 structure. K112 further differs in the linkage between K units formed by the Wzy polymerase, and a different *wzy* gene is found in KL112. However, though both KL106 and KL112 share the *atr8* acetyltransferase gene with KL83, only K83 is acetylated.

## 1. Introduction

*Acinetobacter baumannii* is one of the leading bacterial agents of difficult-to-treat seri- ous nosocomial infections on a global scale. Due to increasing and widespread resistance to carbapenems, one of the last line antibiotics, the World Health Organization listed *A. baumannii* of highest priority for the development of novel therapeutics [1]. However, development of effective alternate therapeutics is made challenging by the highly variable capsular polysaccharide (CPS), which surrounds the *A. baumannii* cell and protects the bacteria from the action of immune system components, as well as disinfectants, desiccation, and certain antimicrobial compounds [2,3,4] and from attack by many phages [5,6,7].

The CPS is a high molecular weight molecule composed of many repeated oligosaccharides (K units). The K units made by different *A. baumannii* strains may differ in sugar composition, sugar linkages, and/or linkages between the K units in the CPS polymer. K units may also be variously decorated with acetyl, acyl, pyruvyl groups, or other moieties (for examples, see [8,9,10,11,12]). This high structural diversity in the species is predominantly due to extensive variation in the genetic content at the chromosomal K locus (KL) that drives CPS biosynthesis [13], and to date, more than 140 KL gene clusters have been identified at this location [14]. Chemical structures are now available for more than 40 different *A. baumannii* CPSs [15], and generally, the resolved CPS structures are consistent with the genes located at the K locus. One aspect of the diversity is that several structures have been found to include sugar substrates that are either only found in *A. baumannii* or otherwise rarely occur in nature; for example, 5,7-di-acetylacinetaminic acid [16], 5,7-di-acetyl-8-epiacinetaminic acid [17], *N*-acetylviosamine [18], and 6-deoxy-l-talose [19].

As further KL gene clusters are found and their corresponding CPS structures are determined, groups of related KL that share genetic features and direct synthesis of related CPS structures have emerged. Previously, a group of eight related KL gene clusters were reported [19], which carry a novel *tle* epimerase gene for the conversion of dTDP-l-rhamnose (dTDP-l-Rha*p*) to dTDP-6-deoxy-l-talose (dTDP-l-6dTal*p*) and *rmlBDAC* genes for the synthesis of dTDP-l-Rha*p*. Five of these gene clusters, KL11, KL83, KL29, KL105, and KL106, shared additional features, differing predominantly from each other in the specific combination of *gtr* glycosyltransferase and *atr* acetyltransferase genes present. The structures reported for K11 and K83 (Figure 1) allowed the encoded enzymes to be assigned to the formation of specific linkages based on shared features identified. How-ever, the K29, K105, and K106 structures remained to be established.

In this work, we determine the K106 structure from *A. baumannii* isolate 48-1789 and correlate the structural data with the gene clusters of KL106 and related KL. We also identify a novel KL112 gene cluster in *A. baumannii* isolate MAR24, which also belongs to this group, and determine the corresponding K112 structure.

## 2. Results

### 2.1. Characterization of the KL106 and KL112 CPS Biosynthesis Gene Clusters

Whole genome sequences were obtained for *A. baumannii* clinical isolates 48-1789 and MAR24. The K locus in the 48-1789 genome was found to carry the KL106 CPS biosynthesis gene cluster, sharing 98.1% identity (99% sequence coverage) with KL106 from *A. baumannii* isolate 219_ABAU (WGS accession number JVPN01000008.1) described in a previous study [19]. KL106 includes *rmlBDAC* genes for dTDP-l-Rha*p* synthesis and a *tle* epimerase gene to generate dTDP-l-6dTal*p*. A novel but related gene cluster with *rmlBDAC* and *tle* genes was also identified at the K locus in the genome of isolate MAR24, and was designated as KL112.

KL106 and KL112 (Figure 2) share most of the genes present, differing only in the region containing *wzx* and *wzy* genes, with an additional *gtr* gene (*gtr183*) found in KL112. Both KL106 and KL112 share a portion of the central region (*gtr27-gtr60-atr8-tle-gtr29-itrA3*) with the previously reported KL83 gene cluster (Figure 2). Indeed, KL106 differs from KL83 only in the presence of a different *wzx* gene and the presence of an additional *gtr* in KL83. Thus, both K106 and K112 K-unit structures are predicted to include the same α-d-GlcpNAc-(1→2)-β-d-Glcp-(1→3)-α-l-6dTalp-(1→3)-β-d-GlcpNAc tetrasaccharide segment as K83 (Figure 1) that is generated by the shared genes. The structure of K106 is also likely to include an identical linkage between the K units due to a shared *wzy* gene, but would lack the Rha side chain found in K83. However, given the unique *wzy* gene and additional *gtr183* gene in KL112, K112 is predicted to include an additional sugar residue with a different linkage between the K units.

### 2.2. Monosaccharide Composition of K106 and K112

Sugar analysis of the CPS preparations from strains 48-1789 and MAR24 by GLC of the acetylated alditols revealed 6dTal, Glc, and GlcNAc in the ratios ~1.0:0.8:3, or Rha, 6dTal, Glc, and GlcNAc in the ratios ~0.1:0.1:0.2:1, respectively (see Materials and Methods and Appendix A). Both d- and l-enantiomers of 6-deoxytalose are not common monosaccharides, though they were found in some bacterial polysaccharides, see additional References in Appendix A. The CPSs were studied by NMR spectroscopy including one-dimensional 1H NMR (see Materials and Methods and Appendix A) and 13C NMR (Figure 3 and Figure 4) and two-dimensional 1H,1H COSY, TOCSY, ROESY, 1H,13C HSQC, and HMBC experiments (see Materials and Methods and Appendix A). Four sugar spin systems including those for β-Glc*p*NAc (unit **A**), α-Glc*p*NAc (unit **D**), α-6dTal*p* (unit **B**), and β-Glc (unit **C**), were identified in the CPS of 48-1789 (Table 1). Spin systems for the same monosaccharides and, in addition, β-Rha*p* (unit E), were found in the CPS of MAR24 (Table 2). All monosaccharides were in the pyranose form. Hence, the additional sugar in K112 is Rha*p*.

### 2.3. The K106 Structure

In the 1H,1H TOCSY spectrum of the CPS of 48-1789, there were correlations for H1 with H2-H6 of unit **A**, H1 with H2 of unit **B**, H1 with H2-H5 of unit **C**, and H1 with H2–H4, H6 of unit **D**. The signals within each spin system were assigned using the 1H,1H COSY spectrum. Relatively large J1,2 coupling constants of 7–8 Hz indicated that units **A** and **C** were β-linked, whereas the α-linked unit **D** was characterized by a smaller J1,2 value (<4 Hz). Comparison of the 13C NMR chemical shifts of unit **B** with published data [20] showed that unit **B** was α-linked. Low-field positions at δ 77.2, 83.3, 76.4, and 80.7 of the signals for C2 of unit **C**, C3 of units **A** and **B**, and C4 of unit **D**, respectively, showed that the CPS is linear and revealed the glycosylation pattern of the monosaccharides.

The sequence of the monosaccharides was determined by the 1H,13C HMBC experiment which showed correlations between the anomeric protons and linked carbons of the neighboring sugar residues including correlations of H1 of unit **A** with C4 unit **D**, H1 of unit **B** with C3 of unit **A**, H1 of unit **C** with C3 of unit **B**, and H1 of unit **D** with C2 of unit **C**. These data also confirmed the substitution pattern in the **K** unit.

Based on these data, it was concluded that the K106 CPS of *A. baumannii* 48-1789 is linear and it has the structure shown in Figure 5.

### 2.4. The K112 Structure

Correlations for units **A**–**D** in the ^1^H,^1^H TOCSY spectrum of the K112 CPS were similar to those in the spectrum of the K106 CPS. In addition, there was a correlation between H1 and H2 of unit **E** and there were no correlations between H1 with H5-H6 for unit **A** and H1 with H6 for unit **D**. Comparison of the _13_C NMR chemical shifts of units **B** and **E** with published data of the corresponding monosaccharides [20,21] showed that unit B was α-linked and unit E was β-linked.

Low-field positions at δ 77.3, 83.0, 76.3, 76.2, and 76.5 of the signals for C2 of unit **C**, C3 of units **A**, **B**, and **D**, and C4 of unit **D**, respectively, showed that the CPS is branched, with four monosaccharide residues (**A**–**D**) in the main chain, 3,4-disubstituted unit **D** at the branching point and unit **E** attached as a side-chain (Figure 5).

The order of the monosaccharides in the K112 CPS of *A. baumannii* MAR24 was determined and the substitution pattern in the K unit was confirmed by the 1H,13C HMBC experiment, which showed the following correlations of the anomeric protons with the linked carbons of the neighboring sugar residues: H1 of unit **A** with C3 of unit **D**, H1 of unit **B** with C3 of unit **A**, H1 of unit C with C3 of unit **B**, H1 of unit **D** with C2 of unit **C**, and H1 of unit **E** with C4 of unit **D**.

The CPS structures established by NMR spectroscopy were corroborated by Smith degradation followed by identification of the resulting oligosaccharides (OS1 for K106 and OS2 and OS3 for K112) by NMR spectroscopy as described above for the CPS (Table 1 and Table 2) and high-resolution electrospray ionization mass spectrometry (HR ESI MS). OS1, *m*/*z* 665.2375; calcd. for C_25_H_42_N2O_17_Na^+^ *m*/*z* 665.2376; OS1 (hydrated form) *m*/*z* 683.2416, calcd. for C_25_H_44_N_2_O_18_Na^+^ *m*/*z* 683.2481; OS2, *m*/*z* 665.2376, calcd. for C25H42N2O17Na^+^ *m*/*z* 665.2376, OS2 (hydrated form) *m*/*z* 683.2481, calcd. for C25H44N2O18Na^+^ *m*/*z* 683.2481; *m*/*z* 641.2417, calcd. for C25H41N2O17^−^ *m*/*z* 641.2411; OS3: *m*/*z* 739.2748, calcd. for C_28_H_48_N_2_O_19_Na^+^ *m*/*z* 739.2743; *m*/*z* 715.2786, calcd. for C_28_H_47_N_2_O_19_^−^ *m*/*z* 715.2779. OS1 and OS2 were found to be the expected products containing glyceraldehyde (C’) as an aglycone that was derived from the 2-substituted β-Glc residue (unit C) (Figure 5). OS3 was suggested to have a cyclic aglycone (C’’) due to incomplete hydrolysis in the destroyed β-Glc residue (unit **C**).

Therefore, the K112 CPS of *A baumannii* MAR24 has the structure presented in Figure 5.

### 2.5. Assignment of Glycosyltransferase Functions

The K106 and K112 structures (Figure 5) both include the same α-d-Glcp*N*Ac-(1→2)-β-d-Glcp-(1→3)-α-l-6dTalp-(1→3)-d-Glcp*N*Ac tetrasaccharide segment as K83, as predicted above from the shared genetic content of the respective KL. The composition of this shared main chain is consistent with the roles predicted previously [19] for the Gtr27, Gtr60, Gtr29, and ItrA3 enzymes in its biosynthesis. Knowing that each unit begins with a β-linked D-Glcp*N*Ac residue transferred by ItrA3 enables a confident assignment of the linkages formed by the various Wzy polymerases. Wzy_K106_ (GenPept accession number QBM04789.1) and Wzy_K83_ (GenPept accession number AHB32311.1) are 80% identical and form the same β-d-Glc*p*NAc-(1→4)-d-Glc*p*NAc linkage between units. Wzy_K112_ (GenPept accession number QNR01097.1) is not significantly related to either Wzy_K106_ or Wzy_K83_, and this is consistent with the β-d-Glcp*N*Ac-(1→3)-d-Glcp*N*Ac linkage between units in the K112 structure is different.

In addition to a shared main chain, K112 includes an l-Rha*p* side branch, like K83, that is attached to the terminal d-Glc*p*NAc residue in the respective main chains. In K112, the remaining Gtr183_K112_ enzyme (GenPept accession number QNR01095.1) would add this sugar via a β-(1→4) linkage, whereas in K83, this sugar is linked by Gtr154K83 (GenPept accession number AHB32312.1) via α-(1→3) linkage. Finally, the l-6dTalp residue in K83 is 2-*O*-acetylated but, although KL106 and KL112 carry the same *atr8* gene as KL83, neither the K106 or K112 structures include an *O*-acetyl group.

## 3. Discussion

The *A. baumannii* K106 and K112 structures elucidated in this study are closely related to the *A. baumannii* K11 and K83 structures reported previously [19]. An unusual feature of this group of related CPS structures is the presence of l-6dTalp, which is a rare sugar component of polysaccharides produced by bacterial species (Bacterial Carbohydrate Structure Database at http://csdb.glycoscience.ru/bacterial/ (accessed on 15 April 2021), see also References in Appendix A). The corresponding CPS gene clusters share genes for synthesis of dTDP-l-Rha*p* and dTDP-l-6dTal, as well as the ItrA3 initial glycosylphosphotransfer to build the lipid-linked β-D-Glc*p*NAc **A**-PP-Und precursor and the glycosyltransferases required for the assembly of the **D**-**B** fragment. Previously, KL106 was also found to be 95% identical to the KL gene cluster from *A. nosocomialis* M2 [19] and deletion of *gtr* genes in this cluster confirmed the order of function [22], which is identical to the assignments made in this study. Hence, the M2 CPS structure, which was not determined previously, would likely be the same as the K106 structure. Though K11 includes the same α-D-Glc*p*NAc-(1→2)-β-d-Glc*p*-(1→3)-α-l-6dTal*p*-(1→3)-β-d-Glc*p*NAc tetrasaccharide fragment (Figure 1), the genetic arrangement of the KL11 gene cluster includes *gtr28-atr6* in place of *gtr60-atr8* that is present in KL106, KL112, and KL83. Gtr28 was previously reported to be 46% identical to Gtr60, and the two glycosyltransferases appear to catalyze formation of the same linkage.

Though structurally and genetically related, these four CPSs differ in the presence or absence, and/or mode of attachment of the α-l-Rha residue (unit **E**). This additional l-Rha residue is either incorporated into the main chain to give a linear trisaccharide fragment (K11), or is present as a side chain that is α-(1→3)-linked (K83) or β-(1→4)-linked (K112) to α-d-Glc*p*NAc. The corresponding gene clusters include genes for different rhamanosyl- transferases, Gtr26_K11_, Gtr154_K83_, and Gtr183_K112_, respectively. Further differences between the four CPS structures are due to the occurrence of different Wzy polymerase genes, which create the connection of the K units by a β-d-Glc*p*NAc (1→3)-α-l-Rha*p* (K11a), β-d-Glc*p*NAc-(1→3)-α-d-Glc*p*NAc (K112), β-d-Glc*p*NAc-(1→4)-α-D-Glc*p*NAc (K106), or β-d-Glc*p*NAc-(1→4)-α-d-Glc*p*NAc (K83) linkage.

An additional difference between structures in this related group is the presence or absence of an *O*-acetyl group at the α-l-6dTalp residue. Each of the four gene clusters includes an acetyltranferase gene adjacent to *tle*: *atr6* in KL11 and *atr8* in KL83, KL106, and KL112. Previously, Atr8 was predicted to be responsible for 2-*O*-acetylation of α-l-6dTal*p* in K83. However, as no acetyl group was found in the K11 structure, a role for Atr6 in CPS modification was not established [19]. In this study, both K106 and K112 lack *O*-acetylation like K11, suggesting either that *atr8* may be inactive in these strains or that the *atr* gene responsible for acetylation of K83 resides elsewhere. Further work will be needed to confirm the differential expression and/or activity of these acetyltransferases, or if an acetyltransferase is encoded elsewhere in the genome as has been observed for other strains [12].

Previous pairwise comparison of gene clusters within this l-6dTal*p-*containing group revealed that the KL11 and KL29 gene clusters, and also the KL83 and KL105 gene clusters, are gene cluster ‘pairs’, differing only in the segment including either *gtr28*/*atr6* or *gtr60*/*atr8*. Thus, the structures of the CPSs of these pairs are expected to be identical, with a potential difference only in the type of *O*-acetylation pattern present. However, due to the possible differential activity of Atr6 and Atr8, this study reinforces the necessity for elucidating structural data rather than forming conclusions about structure based on sequence alone.

## 4. Materials and Methods

### 4.1. Bacterial Strains, Cultivation, and Isolation of CPSs

*A. baumannii* strains 48-1789 and MAR24 were from the collection of multidrug-resistant and extensively drug-resistant *A. baumannii* isolates of the Institute of Antimicrobial Chemotherapy, Smolensk State Medical University (Smolensk, Russia). Bacteria were cultivated in 2 × TY media overnight; cells were harvested by centrifugation (10,000× *g*, 15 min).

Bacterial cells (~1 g) were extracted with 45% aqueous phenol, the extract was dialyzed against distilled water without layer separation and freed from insoluble contaminations by centrifugation. The resultant solution was treated with cold (4 °C) aq. 50% CCl3CO2H; after centrifugation, the supernatant was dialyzed against distilled water and freeze-dried. Crude CPS samples were heated with 2% aqueous AcOH at 100 °C for 2 h, and the purified high-molecular mass CPSs preparations (8 mg from 48-1789 and 20 mg from MAR24) were isolated by gel-permeation chromatography on a column (60 × 2.5 cm) of Sephadex G50 Superfine in 0.1% aqueous AcOH monitored using a differential refrac- tometer (Knauer, Berlin, Germany).

### 4.2. Monosaccharide Analysis

CPS samples (1 mg each) from 48-1789 and MAR24 were hydrolyzed with 2 M TFA (120 °C, 2 h). Monosaccharides were analyzed by GLC of the alditol acetates on a Maestro chromatograph (Interlab, Moscow, Russia) equipped with an HP-5 column (0.32 mm × 30 m) using a temperature program of 160 °C (1 min) to 290 °C at 7 °C min^−1^.

### 4.3. Smith Degradation

CPS samples from 48-1789 (6 mg) and MAR24 (16.3 mg) were oxidized with aq. 0.05M NaIO4 (1 mL) at 20 °C for 48 h in the dark, reduced with NaBH_4_ (12 and 40 mg, respectively) at 20 °C for 16 h. The excess of NaBH4 was destroyed with concentrated AcOH, the solutions were evaporated, and the residues were evaporated with methanol (3 × 1 mL), dissolved in 0.5 mL water, and applied to a column (35 × 2 cm) of TSK HW-40. The modified polysaccharides were eluted with aqueous AcOH and hydrolyzed with 2% AcOH (100 °C, 2 h). Fractionation of the products by gel-permeation chromatography on a column (108 × 1.2 cm) of TSK HW-40 in water gave oligosaccharides OS1 (0.8 mg from 48-1789) and a mixture of OS2 and OS3 (2.3 mg from MAR24) (Figure 5).

### 4.4. NMR Spectroscopy

Samples were deuterium-exchanged by freeze-drying from 99.9% D2O and then examined as solutions in 99.95% D2O. NMR spectra were recorded on a Bruker Avance II 600 MHz spectrometer (Bremen, Germany) at 60 °C. Sodium 3-trimethylsilylpropanoate-2,2,3,3-d4 (δH 0, δC −1.6) was used as internal reference for calibration. Then, 2D NMR spectra were obtained using standard Bruker software, and Bruker TopSpin 2.1 program was used to acquire and process the NMR data. Further, 60-ms MLEV-17 spin-lock time and 150-ms mixing time were used in TOCSY and ROESY experiments, respectively. A 60-ms delay was used for evolution of long-range couplings to optimize HMBC experiments for the coupling constant of JH,C 8 Hz. 1H and 13C NMR chemical shifts of the CPSs and OS1-OS3 are tabulated in Table 1 and Table 2.

### 4.5. Mass Spectrometry

High-resolution electrospray ionization (HR ESI) mass spectrometry was performed in positive and negative ion modes using micrOTOF II and maXis instruments (Bruker Daltonics, Bremen, Germany). Oligosaccharide samples (~50 ng L^−1^) were dissolved in a 1:1 (*v/v*) water–acetonitrile mixture and injected with a syringe at a flow rate of 3 μL min^−1^. Capillary entrance voltage was set at −4500 V (positive ion mode) or 4000 V (negative ion mode). Interface temperature was set at 180 or 200 °C. Nitrogen was used as a drying and nebulizing gas. Mass spectra were acquired in a range from *m*/*z* 50 to *m*/*z* 3000. Internal or external calibration was done with ESI Calibrant Solution (Agilent, Santa Clara, CA, USA). 

### 4.6. Sequencing and Bioinformatic Analysis

Whole genome sequences for *A. baumannii* 48-1789 and MAR-303 isolates were obtained using a Nextera DNA library preparation kit (Illumina, San Diego, CA, USA) and MiSeq platform. Assembly of the short read sequence data was performed with SPAdes v. 3.10 [23]. The K locus sequence located between *fkp*A and *lld*P was extracted and then subjected to KL typing using the Kaptive search tool [14]. Sequence arrangements that could not be identified in the existing *A. baumannii* KL sequence database were assigned a KL number and annotated following the established nomenclature scheme for *A. baumannii* [13]. Fully annotated sequences of KL106 and KL112 were deposited to GenBank under accession numbers MK399430.1 and MT152376.1, respectively. Putative functions of encoded proteins were assigned using BLASTp, as well as searches with the Pfam and CAZy databases.

## Figures and Tables

**Figure 1 ijms-22-05641-f001:**
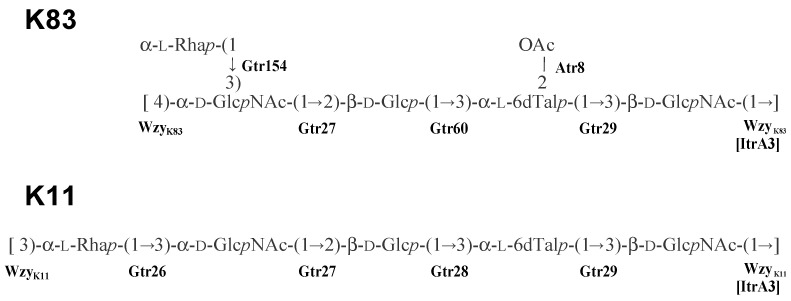
*A. baumannii* K11 and K83 CPS structures established previously [19]. Glycosyltransferases are indicated next to the linkage they were assigned to.

**Figure 2 ijms-22-05641-f002:**
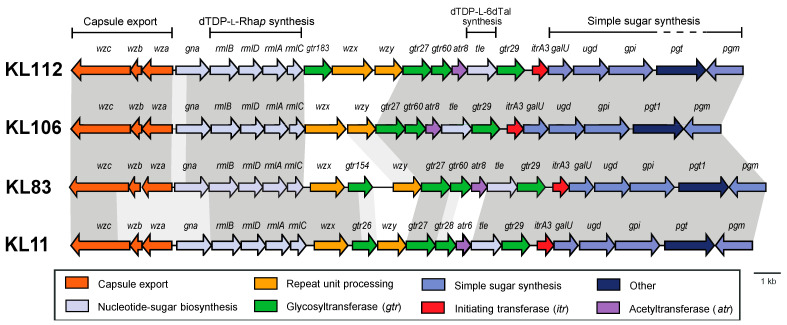
Comparison of the KL106 and KL112 with related *A. baumannii* CPS biosynthesis gene clusters containing the *tle* gene. Dark grey shading between gene clusters shows regions of >97% identity, whereas light shading is regions of 75–97% identity. Figure is drawn to scale from GenBank or WGS accession numbers MT152376.1 (KL112), MK399430.1 (KL106), KC526898.2 (KL83), and KF002790.2 (KL11). The scale bar and color scheme are shown below.

**Figure 3 ijms-22-05641-f003:**
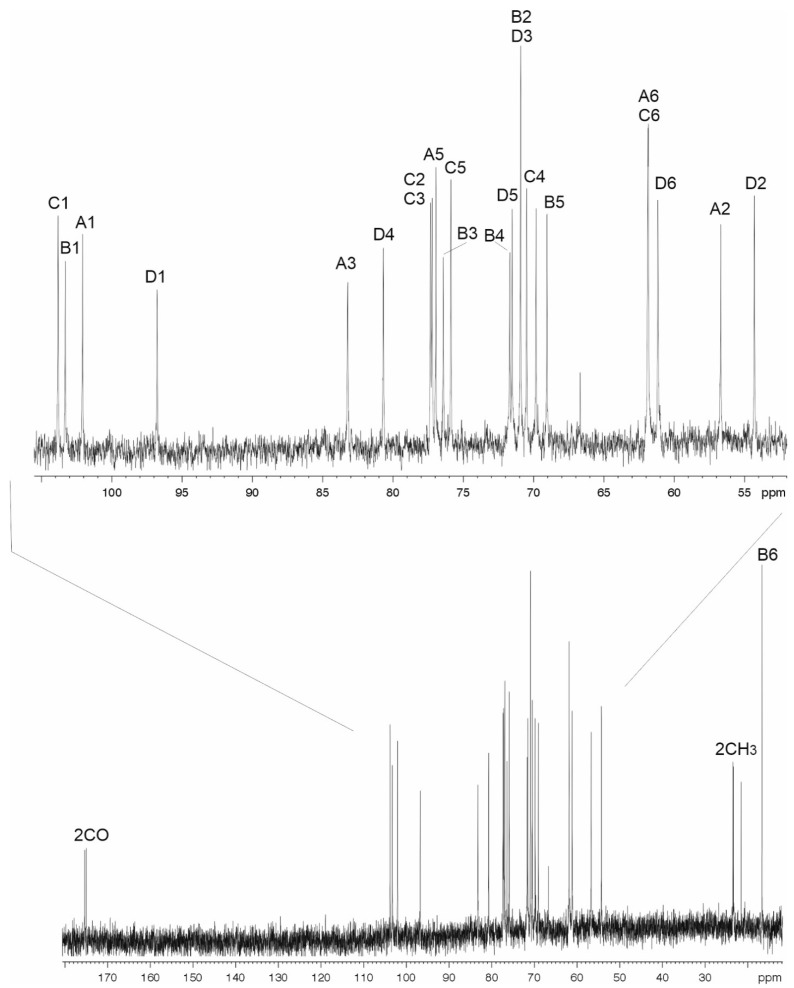
13C NMR spectrum of the CPS of *A. baumanii* K106 (48-1789).

**Figure 4 ijms-22-05641-f004:**
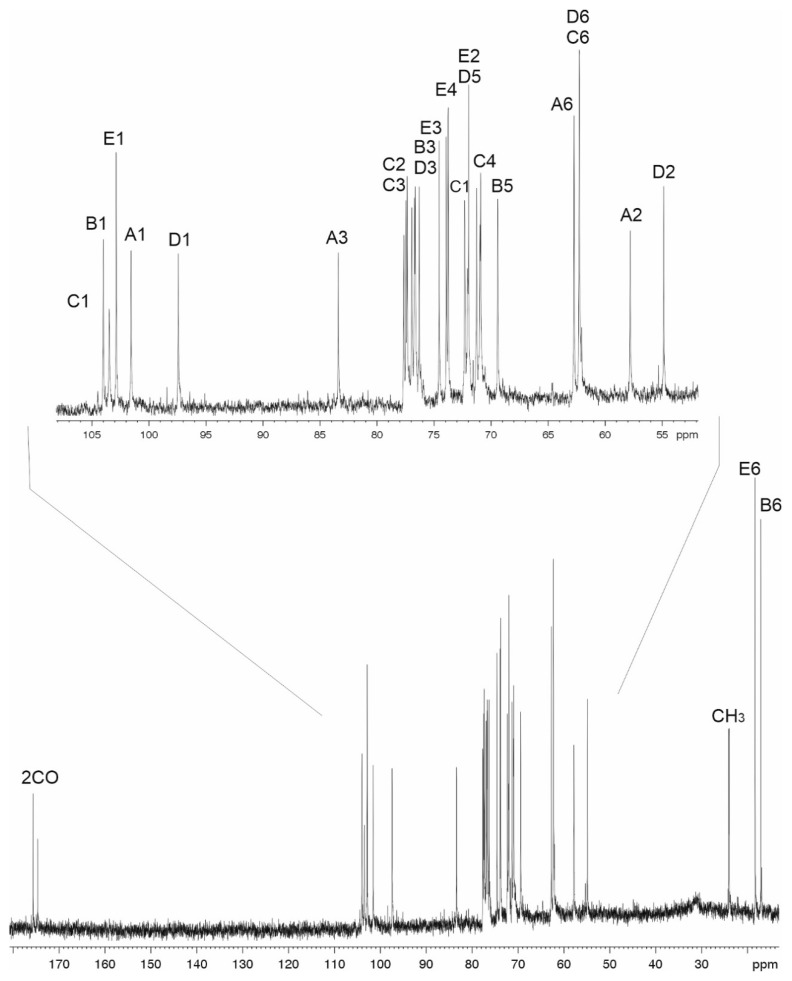
13C NMR spectrum of the CPS of *A. baumanii* K112 (MAR24).

**Figure 5 ijms-22-05641-f005:**
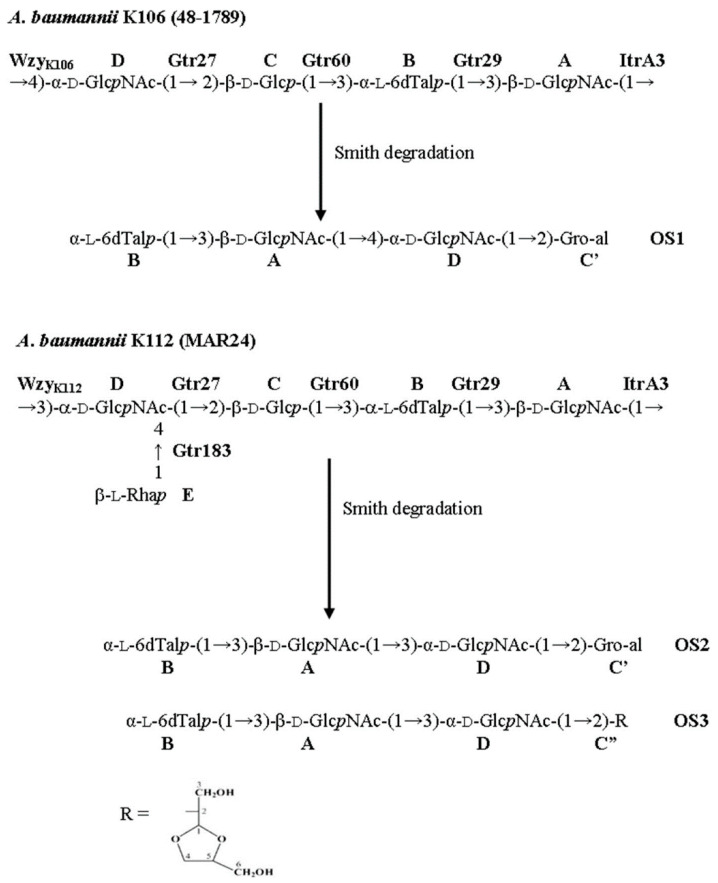
Structures of the capsular polysaccharides (CPSs) of *A. baumannii*, K106 (48-1789), K112 (MAR24), and oligosaccharides (**OS1**—**OS3**) derived by Smith degradation of the CPSs. Gro-al indicates glyceraldehyde. Glycosyltransferases are shown above or next to the linkage each is expected to form.

**Table 1 ijms-22-05641-t001:** ^1^H and ^13^C NMR chemical shifts (δ, ppm) of the K106 capsular polysaccharide of *A. baumannii* 48-1789 and oligosaccharide OS1 derived by Smith degradation.

Sugar Residue	C-1 *H-1*	C-2 *H-2*	C-3 *H-3*	C-4 *H-4*	C-5 *H-5*	C-6 *H-6 (6a*,*6b)*
CPS ^a^						
→3)-β-d-Glc*p*NAc-(1→ **A**	102.1	56.6	83.3	69.7	77.3	61.9
	4.65	*3.85*	*3.65*	*3.49*	*3.51*	*3.78*, *3.93*
→3)-α-l-6dTal*p*-(1→ **B**	103.3	70.976.4	71.6	69.0	16.7	
	*4.98*	*3.88*	*3.86*	*3.66*	*4.23*	*1.22*
→2)-β-d-Glc*p*-(1→ **C**	103.8	77.2	75.9	70.6	77.3	61.8
	*4.73*	*3.51*	*3.57*	*3.48*	*3.51*	*3.78*, *3.93*
→4)-α-d-Glc*p*NAc-(1→ **D**	96.8	54.3	71.0	80.7	71.5	61.2
	*5.45*	*3.96*	*3.91*	*3.68*	*4.12*	*3.67*, *3.81*
OS1 ^b^						
α-l-6dTal*p*-(1→ **B**	103.4	71.7	66.9	73.4	69.1	16.8
	*4.99*	*3.69*	*3.86*	*3.75*	*4.27*	*1.22*
→3)-β-d-Glc*p*NAc-(1→ A	102.1	56.8	82.7	69.9	77.3	62.0
	*4.67*	*3.86*	*3.68*	*3.53*	*3.53*	*3.76*, *3.94*
α-d-Glc*p*NAc-(1→ **D**	98.1	54.6	70.8	81.0	78.1	61.4
	*5.21*	*3.93*	*3.93*	*3.67*	*4.28*	*3.69*, *3.80*
→2)-Gro-al **C’**	90.8	74.9	62.3			
	*5.10*	*3.60*	*3.77*			

^1^H NMR chemical shifts are italicized. Gro-al indicates glyceraldehyde, which is present in the hydrated form. Chemical shifts of the *N*-acetyl groups are ^a^ δH 1.98-2.08, δC 23.3-23.5(Me) and 175.0 and 175.4 (CO); ^b^ δH 2.05-2.09, δC 23.3-23.4 (Me).

**Table 2 ijms-22-05641-t002:** 1H and 13C NMR chemical shifts (δ, ppm) of the K112 capsular polysaccharide of *A. baumanii* MAR24 and oligosaccharides OS2 and OS3 derived by Smith degradation.

Sugar Residue		C-1 *H-1*	C-2 *H-2*	C-3 *H-3*	C-4 *H-4*	C-5 *H-5*	C-6 *H-6 (6a*,*6b)*
CPS ^a^							
→3)-β-d-Glc*p*NAc-(1→ **A**		101.1	57.4	83.0	70.6	76.9	62.7
		*4.61*	*3.67*	*3.71*	*3.41*	*3.42*	*3.76*, *3.97*
→3)-α-l-6dTal*p*-(1→ **B**		103.1	70.9	76.3	71.7	69.0	16.7
		*5.00*	*3.88*	*3.83*	*3.64*	*4.22*	*1.22*
→2)-β-d-Glc*p*-(1→ **C**		103.6	77.3	75.9	70.5	76.9	62.3
		*4.73*	*3.52*	*3.57*	*3.48*	*3.44*	*3.83*, *3.88*
→3,4)-α-d-Glc*p*NAc-(1→ **D**		97.0	54.5	76.2	76.5	71.9	62.2
		*5.38*	*4.05*	*4.22*	*3.74*	*4.13*	*3.76*, *3.85*
β-l-Rha*p*-(1→ **E**		102.5	71.5	74.2	73.4	73.6	17.9
		*4.86*	*4.19*	*3.57*	*3.39*	*3.42*	*1.31*
OS2, OS3 ^b^							
α-l-6dTal*p*-(1→ **B**		103.2	71.6	66.7	73.4	68.9	16.7
		*4.98*	*3.68*	*3.86*	*3.74*	*4.27*	*1.19*
→3)-β-d-Glc*p*NAc-(1→ **A** ^d^		101.6	56.9	82.4	69.7	76.9	62.2
		*4.66*	*3.79*	*3.70*	*3.53*	*3.49*	*3.78*, *3.82*
→3)-α-d-Glc*p*NAc-(1→ **D** ^d^		99.3 ^c^, 98.4	53.9	80.5	69.7	73.0	61.8
		*5.18*^c^, *5.08*	*4.01*	*3.95*	*3.58*	*3.91*	*3.84*, *3.93*
→2)-Gro-al **C’**		90.5	82.2	67.4			
		*5.11*	*3.60*	*3.84*,*4.03*			
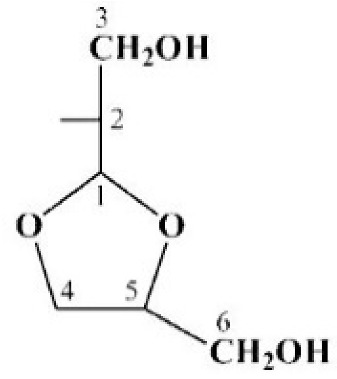	**C”**	104.2 *5.06*	79.0 *3.78*	67.4 *3.84*, *4.03*	62.8 *3.62*, *3.72*	77.9 *4.28*	61.7 *not found*

1H NMR chemical shifts are italicized. Gro-al indicates glyceraldehyde, which is present in the hydrated form. Chemical shifts of the *N*-acetyl group are ^a^ δ_H_ 2.00, 2.08; δ_C_ 23.6 (Me) and 174.2, 175.2 (CO); ^d^ δ_H_ 2.01, 2.07; δ_C_ 23.5 (Me) and 169.9, 175.3 (CO). ^b^ Obtained as a mixture; C’’ relates to OS3. ^c^ Chemical shifts for OS 3.

## Data Availability

GenBank accession numbers: MK399430.1 (KL106); MT152376.1 (KL112).

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
