# Peer review of "Acinetobacter baumannii K106 and K112: Two Structurally and Genetically Related 6-Deoxy-l-talose-Containing Capsular Polysaccharides"

_ijms, 2021, doi:10.3390/ijms22115641_

Round 1
Reviewer 1 Report
Acinetobacter baumannii strains are of considerable biomedical importance as these bacteria have become resistant against many antibiotics leading to serious and occasionally fatal infections in intensive care units. Capsular polysaccharides serve as a main bacterial protective barrier and detailed structural knowledge of these polymers and their underlying genetic basis is thus mandatory.
Extending previous studies, the authors present the structure of two related K-antigens containing the rare sugar 6-deoxy-L-talose. The paper is concisely written with convincing NMR data supported by oligosaccharide fragments obtained by Smith degradation.
Only minor amendments are suggested:
Line 46: …by many phages [5-
Line 51: predominantely (also line 66)
Lines 58 & 59: replace …acinetominic by ….acinetaminic acid (as it corresponds to an amino sugar)
Correct also typos for diacetyl
Line 77: are indicated next….
Table 2: The last 3 rows showed by reformatted to improve legibility
Line 164: ..2-substituted
Figure 3: Two signals have been denoted as A3 (the upfield shifted one should be labeled as D4); labels for A5 and B3 could also be added
References: a few additional references for the occurrence of 6-deoxy-L-talose might be added
Author Response
Only minor amendments are suggested:
Line 46: …by many phages [5-
- Corrected.
Line 51: "predominantely" should be "predominantly" (also line 66)
- Corrected (both).
Lines 58 & 59: replace …acinetominic by ….acinetaminic acid (as it corresponds to an amino sugar)
- Replaced.
Correct also typos for diacetyl
- Corrected.
Line 77: are indicated next…
- Corrected.
Table 2: The last 3 rows showed by reformatted to improve legibility
- Reformatted.
Line 164: ...2-substituted
- Corrected
Figure 3: Two signals have been denoted as A3 (the upfield shifted one should be labeled as D4); labels for A5 and B3 could also be added
- Corrected
References: a few additional references for the occurrence of 6-deoxy-L-talose might be added
- 6-Deoxy-L-talose cannot be regarded as a common sugar, nevertheless, it was found in some bacteria (as well as its D enantiomer); in almost dozen of papers these carbohydrates (L or D) are mentioned. Because of these papers are not related to CPS (or any other PS) of Acinetobacter baumannii, we placed the additional list in Supplementary Materials.
Reviewer 2 Report
The article by Anastasiya A. Kasimova et al. presents a successful merge of two remotely related techniques – classical structural analysis of capsular antigens and the underlying genetic data. It is vital for our understanding of the ever changing microorganisms to identify the specific antigenic structures of pathogenic bacteria, so new approaches of tackling the pathogen resistance to antibiotics and other escape routes can be devised. Such is the case of the presented structures of the capsular polysaccharides of Acinetobacter baumanii. More importantly, the Authors are aware that the genotypic data alone is not sufficient for a complete characterization of the resulting fenotypes. I have no major concerns, as the manuscript is properly structured and provides the necessary data to support the identified CPS. Some spotted minor issues are mostly editorial and include:
1) For the clarity, the readability of tables should be improved (e.g. see to the formatting of the chemical shifts of residue A in Table 1, but also residues C’ & C’’ in Table 2);
2) Page 5, lines 153-158 – the description of the connectivity between residues D and C is missing in the text;
3) In the 13C spectra (Figures 3 & 4) - the „NAc” resonances should be indicated as originating from the carbonyl- and methyl- of NAc groups,
4) Two carbonyl carbon signals in Figure 4 can be clearly seen and just a single resonance in Figure 3, but in both CPS there are two GlcNAc residues. Please explain briefly.
5) The abbreviations used for the acetic acid are inconsistent (e.g. compare lines 256 and 270)
Author Response
1) For the clarity, the readability of tables should be improved (e.g. see to the formatting of the chemical shifts of residue A in Table 1, but also residues C’ & C’’ in Table 2);
- Corrected.
2) Page 5, lines 153-158 – the description of the connectivity between residues D and C is missing in the text;
- Corrected.
3) In the 13C spectra (Figures 3 & 4) - the „NAc” resonances should be indicated as originating from the carbonyl- and methyl- of NAc groups,
- Corrected.
4) Two carbonyl carbon signals in Figure 4 can be clearly seen and just a single resonance in Figure 3, but in both CPS there are two GlcNAc residues. Please explain briefly.
- Indeed, there are two very close carbonyl peaks in Figure 3 (48-1789), for MAR-24, carbonyl resonances are separated by ca. 1 ppm., possibly, due to substitution of residue D with Rhap (downfield shift).
5) The abbreviations used for the acetic acid are inconsistent (e.g. compare lines 256 and 270)
- Corrected.
Reviewer 3 Report
The present paper reports the structural study and genetic characterization of two capsular polysaccharides from Acinetobacter baumannii. The structures were characterized by appropriate methods, and the conclusions were confirmed.
Minor comments:
- The figures and analysis of GC results were suggested to be supplemented.
- The whole spectra of 1H, COSY, TOCSY, ROESY, HSQC, and HMBC were suggested to be provided.
- Please rename paragraph 2.2.
- Please place tables and spectra in paragraphs referring to polysaccharides.
Author Response
Minor comments:
- The figures and analysis of GC results were suggested to be supplemented.
- GC is added to Supplementary Materials
- The whole spectra of 1H, COSY, TOCSY, ROESY, HSQC, and HMBC were suggested to be provided.
- 2D NMR spectra are added to Supplementary Materials.
- Please rename paragraph 2.2.
- 2.2. Monosaccharide composition of K106 and K112.
- Please place tables and spectra in paragraphs referring to polysaccharides.
- Replaced as required.
Reviewer 4 Report
Dear Editor,
I read the manuscript by Kasimova et al., entitled: "Acinetobacter baumannii K106 and K112: Two structurally and genetically related 6-deoxy-L-talose-containing capsular polysaccharides ". The manuscript describes the primary structure of K106 and K112 capsular polysaccharides as well as their related gene clusters. The experiments were well perform, the results are sound and well explained. I have only minor suggestions.
line 58: acetyl (twice)
legend to figure 1: indicated
line 114: All monosaccharides were in the pyranose form.
line 178: biosynthesis
lines 183-185: WzyK112 (GenPept accession number QNR01097.1) is not significantly related to either WzyK106 or WzyK83, in agreement with the related catalyzed beta (1-3) linkage between the D-GlcpNAc and D-GlcpNAc residues in the K112 structure.
Line 196: Glycosyltransferases are reported above each linkage they are expected to form.
Lines 267-275: the authors perform Smith degradation on the samples, but they obtained a glyceraldehyde at the reducing end or a cyclic derivative. Can the authors explain why they obtained the aldehyde in place of glycerol which should have formed instead after reduction?
Lines 295-300: HR ESI MS. In my opinion, these are results and should be moved into the results paragraph.
Author Response
line 58: acetyl (twice)
- Corrected.
legend to figure 1: indicated
- Corrected.
line 114: Should be: "All monosaccharides were in the pyranose form."
- Corrected.
line 178: biosynthesis
- Corrected.
lines 183-185: WzyK112 (GenPept accession number QNR01097.1) is not significantly related to either WzyK106 or WzyK83, in agreement with the related catalyzed beta (1-3) linkage between the D-GlcpNAc and D-GlcpNAc residues in the K112 structure.
- Corrected: "...this is consistent with..."
Line 196: Glycosyltransferases are reported above each linkage they are expected to form.
- Corrected: "...above or next..."
Lines 267-275: the authors perform Smith degradation on the samples, but they obtained a glyceraldehyde at the reducing end or a cyclic derivative. Can the authors explain why they obtained the aldehyde in place of glycerol which should have formed instead after reduction?
- This is correct: Smith-generated polymeric hemiacetals were subjected to mild acid hydrolysis, and no repeated reduction was done. To obtain glycosyl glycerols, the second borohydride reduction is needed (or hydrolysis should be done just after periodate cleavage).
Lines 295-300: HR ESI MS. In my opinion, these are results and should be moved into the results paragraph.
HR ESI MS description was transferred to "Results".